

# Effects of spatial soil moisture variability in forests plots on simulated groundwater recharge estimates

Thomas Fichtner[1]; Yuly Juliana Aguilar Avila[1]; Andreas Hartmann[1], Stefan Seeger[2], Martin Maier[2], Stephan Raspe[3];

[1]Institute of Groundwater Management Technische Universität Dresden, Dresden, 01069, Germany
[2]Georg-August-Universität Göttingen, Soil Physics, Göttingen, 37077, Germany
[3]Bayerische Landesanstalt für Wald und Forstwirtschaft, Department soil and climate, Freising, 85354, Germany

*Correspondence to*: Thomas Fichtner (thomas.fichtner@tu-dresden.de)

**Abstract.**

Soil-Vegetation-Atmosphere Transfer (SVAT) models are essential tools for simulating and underploting the dynamic interactions governing water balance components within forest ecosystems. These models are widely employed to predict hydrological responses to environmental change, including the impacts of shifting meteorological conditions on forested landscapes. Despite their usefulness, the reliability of SVAT models is frequently compromised by uncertainties arising from incomplete or imprecise input data. These limitations often result in model assumptions that may lead to over- or underestimation of critical water balance components such as groundwater recharge. In order to improve the accuracy of SVAT models, observed soil moisture data are integrated to enhance parameterization processes by aligning simulated outputs with measured values. However, uncertainties remain regarding the selection of representative soil moisture profiles for calibration and the extent of measurements necessary to robustly characterize a forest plot. To address these challenges, the present study explores the spatial variability of soil moisture across two forested plots with contrasting soil and vegetation conditions by the deployment of an extensive network of soil moisture probes in 11 profiles per plot. The influence of soil moisture variability on the adjustment of model input parameters during the calibration process and its subsequent impact on the computation of groundwater recharge is evaluated. The findings reveal that soil moisture variability at the plot characterized by a heterogeneous soil was greater, both horizontally and in depth, throughout the study period. These patterns of variability are also mirrored in the different parameter sets obtained from the calibration of the LWF Brook90 model, based on the recorded soil moisture time series in each of the 11 profiles per plot. The most significant variation is observed in the infiltration and hydraulic soil parameters, whereby this is more pronounced at the plot with heterogeneous soil structure. Nevertheless, when examining the groundwater recharge rates calculated using the 30 best-performing parameter sets for each of the 11 profiles, both plots exhibited comparable temporal patterns and in particular similar variations in total volumes of groundwater recharge. These results suggest that model-inherent uncertainties, including parameter interactions, equifinality and dimensional simplifications, have a stronger impact on model outputs than uncertainties arising from variability in soil moisture caused by spatial heterogeneity of soil texture and hydraulic properties within the plot. Taking into account both sources of uncertainty,





the application of bootstrapping techniques demonstrated that groundwater recharge could be reliably estimated using data from only 6 to 7 soil profiles per plot, providing a representative picture of its spatial variability. In general, the results indicate that using data from only a few soil profiles is not sufficient to capture the full range of groundwater recharge dynamics.

**Keywords:**        Soil moisture variability, Soil-Vegetation-Atmosphere Transfer model, groundwater recharge

## 1        Introduction

Forests are vital contributors to the hydrological cycle, playing a pivotal role in aquifer recharge while safeguarding the quality and availability of freshwater resources (Chang, 2012). Acting as natural filtration systems, they regulate the movement of

water from the topsoil into groundwater reservoirs. The vegetation in forests, especially trees, intercepts rainfall, consuming water by root water uptake as well as transpiration and allowing it to infiltrate slowly into the soil (Hewlett, 2003). Protecting and managing these forested areas is essential for maintaining sustainable groundwater replenishment in both quantity and quality. Furthermore, the availability of water is a critical determinant for the ecological functionality and long-term viability of forest ecosystems. Forests exhibit a pronounced sensitivity to variability in water supply, with significant implications for

their growth dynamics, resilience to environmental stressors, and overall productivity (Williams et al., 2013). One of the greatest challenges facing forest ecosystems is the alteration of meteorological conditions due to climate change. Shifts in precipitation patterns, including reduced rainfall during the growing season and an increase in extreme weather events such as droughts and heavy precipitation leading to excessive surface runoff, represent critical challenges. These changes constrain water availability for trees and inhibit water fluxes from the unsaturated soil zone to underlying groundwater reserves

(Meusburger et al., 2022). Consequently, significant effects on forest structure and species distribution arise, including stress reactions such as widespread tree mortality, reduced canopy cover and increased susceptibility to pests and diseases (Gebeyehu and Hirpo, 2019; Klesse et al., 2023; Senf et al., 2020). Given these challenges, it is essential to underplot and quantify the processes governing forest water balance and to estimate accurately the volume of water percolating into the ground eventually reaching the groundwater table to become recharge. Hereby, soil water fluxes leading to groundwater recharge are of particular

interest due to their critical role in the hydrological cycle and their influence on subsurface dynamics and long-term water resource sustainability in forested areas.

However, precise estimation of groundwater recharge remains inherently complex as it demands detailed insights into the multifaceted interactions among soil properties, vegetation characteristics, and atmospheric dynamics within forest ecosystems (Schmidt-Walter et al., 2020). In general, key components influencing groundwater recharge in forested landscapes include

precipitation, interception, evaporation, surface runoff, transpiration, percolation, and soil storage capacity. The interplay of these components is regulated by an various site-specific factors, such as climatic conditions (e.g., temperature and precipitation patterns), forest characteristics (e.g., species composition, structural age, plot density, canopy architecture, root system development, and overall tree health), understory vegetation, and soil attributes (e.g., texture, type, and permeability) (Chang, 2006). To address these complexities, environmental monitoring in forest ecosystems seeks to quantify water fluxes



with precision as a basis for determining water availability for transpiration across different tree species and its contribution to groundwater recharge.

Using Soil-Vegetation-Atmosphere Transfer (SVAT) models has been established as indispensable tools for simulating hydrological processes within forest ecosystems as they can effectively capture the temporal dynamics of soil moisture and estimate water fluxes in forest environments (Speich et al. 2020; van der Salm et al. 2007). However, their predictive accuracy

is often constrained by limitations in the quality and availability of input data as well as by the challenges of defining appropriate initial and boundary conditions. A large number of parameters related to canopy structure, vegetation characteristics, root distribution patterns, and soil hydraulic properties must be defined for effective model implementation (Meusburger et al., 2022). Yet, many of these parameters cannot be directly derived from field observations, resulting in significant uncertainties in parameter estimation (Franks et al., 1997). Such uncertainties frequently can lead to over- or

underestimations of critical hydrological components, including water available for transpiration and percolation to the groundwater, thereby reducing the SVAT model's predictive capability (Kirchner, 2006; Kuppel et al., 2018). To address this issue, automatic calibration techniques leveraging observed site-specific soil moisture measurements, are employed. Incorporating these site-specific observations enables refinement of input parameters, effectively reducing mismatches and potentially enhancing the reliability of water balance predictions. This improved parameterization facilitates a more accurate

representation of hydrological dynamics, contributing to a better understanding of water fluxes and their interrelations within forested landscapes.

However, soil moisture exhibits high spatial and temporal variability even within forest plots, driven by factors such as heterogeneities in soil texture, hydraulic properties, topographic gradients, and dynamic interactions with surface water systems, precipitation, and vegetation distribution (Vereecken et al., 2016; Western et al., 2004). Numerous studies have

investigated spatial soil moisture variability (Choi et al. 2007; Fatichi et al. 2015; Mohanty et al. 2000; Ojha et al. 2014; Teuling and Troch 2005; Vereecken et al. 2008; Western et al. 1999), consistently demonstrating that that variability tends to increase across larger spatial scales (Famiglietti et al. 2008; Western et al. 1999). This variability poses a challenge for model calibration, as a parameter set calibrated to a single location often fails to capture the full range of observed soil moisture conditions within a study area (Beven 2006). Recognizing spatial variability in soil moisture is crucial for improving the

predictive performance of hydrological models, particularly in the context of water balance components such as evapotranspiration and infiltration (Famiglietti and Wood 1994). Research has shown that spatial differences in soil hydraulic properties can lead to substantial variations in simulated water balance components, such as transpiration, runoff, and deep percolation (Montzka et al., 2017). However, uncertainty remains regarding the optimal quantity and selection of soil moisture observations necessary to adequately represent a plot for model calibration and estimation of groundwater recharge. In the

context of forest environmental monitoring, the installation of soil moisture observation profiles is often restricted to a limited number of locations (Vorobevskii et al., 2024). This limitation is primarily attributed to the significant financial investment required for advanced measurement technologies, compounded by a general underestimation of the critical role of representative soil moisture data in deriving reliable estimations of forest water balance components.





Recognizing these challenges, this present study seeks to evaluate the gains of installing a larger number of soil moisture
sensors to obtain data for model calibration. To address this, an expanded monitoring network comprising multiple soil
moisture probes was established across two forest plots at different sites in Germany, each characterized by contrasting
environmental and boundary conditions. The collected data were analysed to identify and underplot spatial and temporal soil
moisture variability across the study areas, including differences at distinct depths and locations. The observations were further
used for the calibration of the SVAT model LWF-BROOK90.jl to assess their influence on estimated model parameters and
model outputs, especially on ground water recharge. By addressing uncertainties associated with soil moisture variability and
model parameterization, this our analysis contributes to the ongoing discourse on the spatial resolution required for
hydrological model calibration. The findings should emphasize the importance of balancing single versus multiple
parameterizations to ensure representativeness in heterogeneous forest landscapes, ultimately enhancing the accuracy of
groundwater recharge estimations in forest landscapes.

## 2    Methods

### 2.1    Study sites & data

The research concentrated on two sites within the ICOS (Integrated Carbon Observation System) monitoring network, which
are also part of the IPCC Network  (Intergovernmental Panel on Climate Change), selected for contrasting soil characteristics
and their well-established infrastructure and suitability for comprehensive data collection (Table 1).

**Table 1**   Characterisation of the two sites instrumented as part of the study

|  | **Kienhorst** | **Tharandt** |
|---|---|---|
| **Coordinates (-)** | 52°58' N / 13°39' E | 50°57'N, 13°34' E |
| **Elevation (m)** | 66 | 385 |
| **Slope (°)** | 0 | 7 |
| **Median Temperature (°C)** | 8.5 | 8.2 |
| **Annual Precipitation (mm)** | 577 | 843 |
| **Vegetation** | Pinus / Vaccinium myrtillus / Bryophyta | Picea / Larix/  Bryophyta |
| **Soil type \*, soil texture, stone content** | Haplic Podzol, sand, no stones | Haplic Podzol, silt,  10 – 20%, partially perching properties |
| **Geology** | Glacial sediments including their periglacial overprints | Periglacial sediments consisting of debris from rhyolite and loess |
| **Hydrogeology** | Water level upper aquifer -17 m b.g. on average | Water level upper aquifer -13 m b.g. on average |

(Kallweit and Engel, 2016), (Anchorstation Tharandter Wald - Ökomessfeld), *(IUSS Working Group (WRB), 2022)





### 2.1.1 Study site Kienhorst

The study area is situated in the heart of the Schorfheide, the largest continuous forested region in the state of Brandenburg, Germany (Fig. 1A). This plot consists of a 115-year-old pines, with ground vegetation primarily composed of dwarf shrubs, blueberry herbs and branch mosses (Kallweit and Engel, 2016). Humus form is raw humus with an Of/Oh ratio of 6 (IUSS

Working Group (WRB), 2022). The mineral soil consists of fine sand, and roots seem to grow deeper than 1.5 m (Fig. 1B).

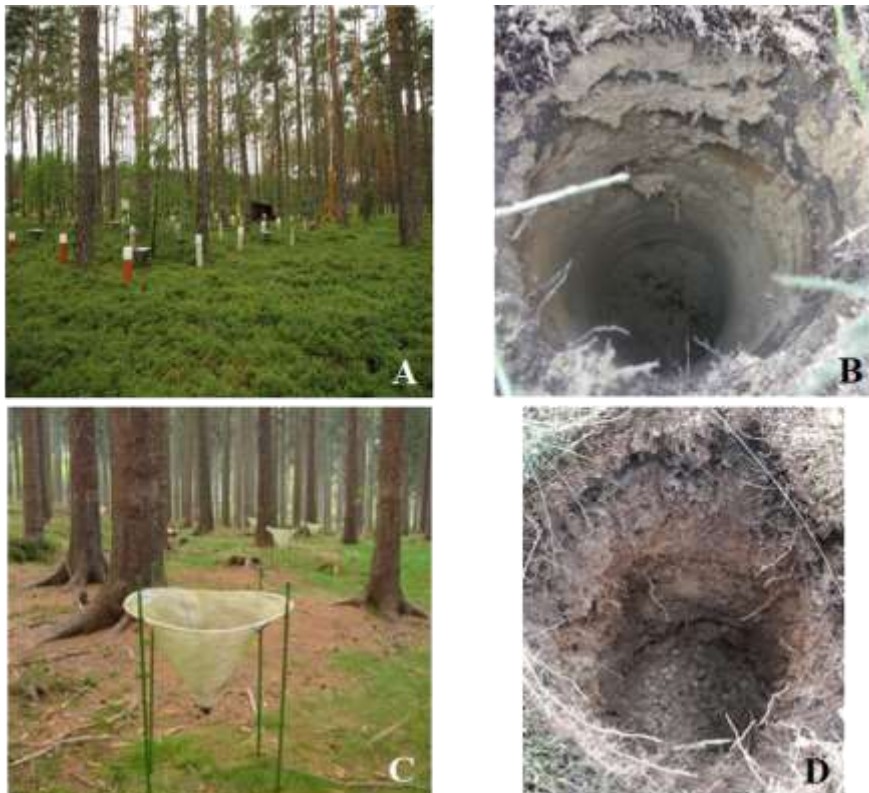

**Figure 1** View of Kienhorst plot (A), soil characteristics of Kienhorst plot (B), view of Tharandt plot (C), soil characteristics of Tharandt plot (D) (photograph by Fichtner, 2023)

### 2.2.2 Study site Tharandt

The study site is situated in the heart of Tharandter Wald, a dense forest covering approximately 6000 hectares on the lower reaches of the northern slopes of the Ore Mountains (Fig. 1C). The forest is characterized by a 129-year-old spruce plot, with ground vegetation primarily composed of grasses and mosses (Anchorstation Tharandter Wald - Ökomessfeld). Humus form is raw humus overlaying a loamy mineral soil with up to 20% stone content throughout the profile (Fig. 1D). Roots seem to grow not deeper than 0.8 m. At different locations on the plot (approx. 35%), the subsoil > 0.5 m depth exhibits redoximorphic

patterns indicating perching properties, which means extremely low permeability. This results in the accumulation of stagnant



water during periods of heavy rainfall, as the infiltrating precipitation encounters significant resistance, hindering its downward movement through the soil layers (Braeutigam, 2012).

## 2.2 Soil moisture measurements

### 2.2.1 Set up soil moisture monitoring network

At the two study sites located in distinct climatic regions of Germany, extensive networks of 44 soil moisture probes were deployed. Each of the 11 soil profiles was equipped with four probes placed at depths of 10, 30, 50, and 80 cm (Fig. 2C). The study utilized SMT100 soil moisture probes (Truebner Company 2025), with integrated temperature measurement, operating based on the Time Domain Transmission (TDT) principle (Fig. 2A) (Qu et al., 2013). The specified accuracy for absolute measurements is 3 vol. % for soil moisture (without site-specific calibration) and between 0.2°C and 0.4°C for soil temperature.


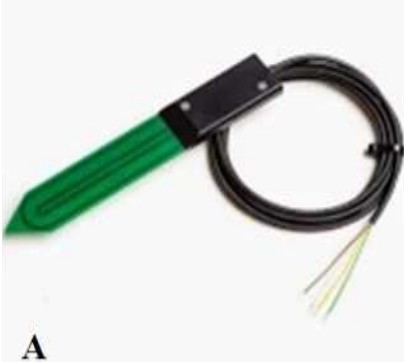 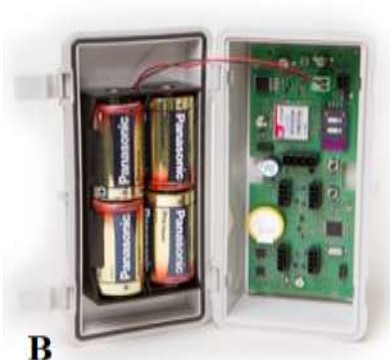 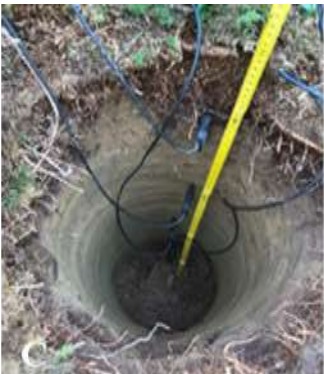

**Figure 2** SMT100 Sensor (Truebner Company)(A), Datalogger TrueLog200 (Truebner Company)(B), soil profile with installed sensors (photograph by Fichtner, 2023) (C)

The sensor provides average measurement values across its full length of 10 cm. In this study, the manufacturer's calibration
was applied instead of site-specific calibration, as the focus was on the soil moisture dynamics rather than absolute values (Sprenger et al. 2015; Demand et al. 2019). To prevent water accumulation on the probes and ensure minimal interference with vertical vapor fluxes, the probes were installed horizontally with their narrow edge oriented vertically (Fig. 2C). Data was collected every 10 minutes using the battery-powered TrueLog200 data logger (Fig. 2B). The data loggers are configured and accessed via the accompanying logger software. Equipped with a GSM modem, the loggers can transmit recorded data through
the mobile phone network.

### 2.2.2 Selection of positioning soil moisture measurements

To identify soil moisture variability at the study plot, the location of the 11 soil profiles per plot were installed at randomized locations within an area of 20 x 20 meters (Fig. 3A+B). Previous research has demonstrated that installing soil moisture sensors



in a minimum of 10 profiles is sufficient to capture plot-specific variability effectively (Berthelin et al., 2020). The random
sampling approach was employed to determine the soil profile coordinates, ensuring that each location was selected with equal
probability. This method was chosen to maintain the independence of data points, facilitating robust statistical analysis.
Additionally, randomized placement across the study area helps to avoid potential systematic errors caused by unrecognized
gradients in soil moisture distribution. This strategy ensures comprehensive coverage of the variability present in the field.

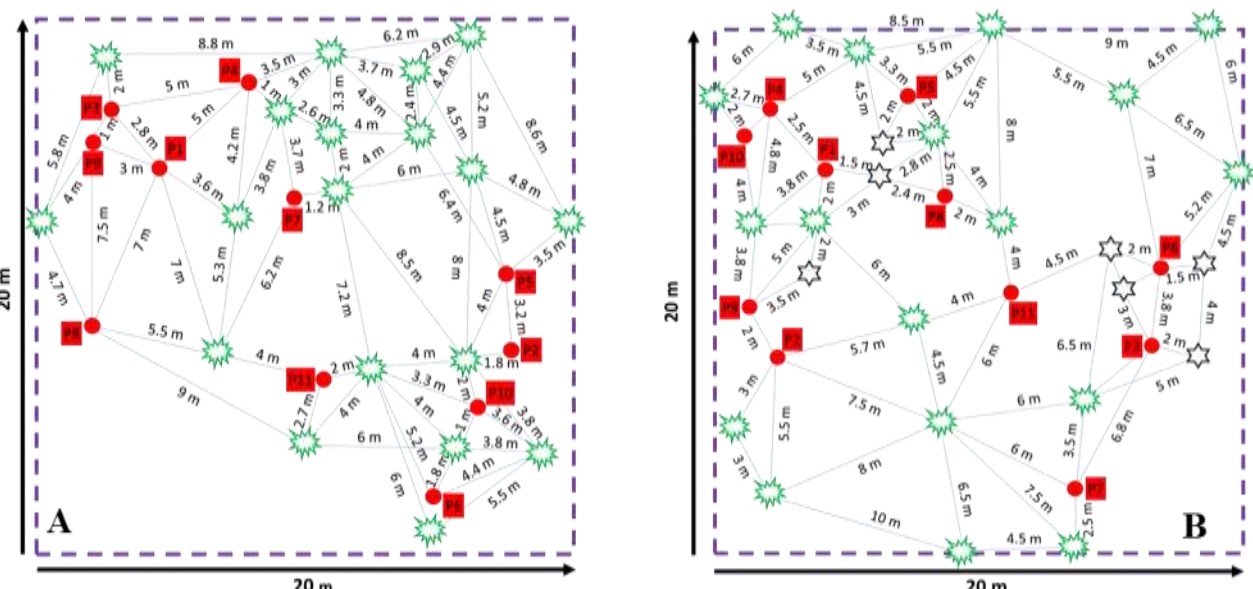

**Figure 3** Randomly distributed soil profiles (red) in the 20 x 20 m plot at Kienhorst site (A) and at Tharandt site (B), living trees = green,
tree stumps = black

## 2.3    Water balance model LWF BROOK90.jl

Water balance models or Soil–vegetation–atmosphere transfer (SVAT) models are valuable tools for estimating detailed
atmosphere–plant–soil water exchange using a 1D simplification representing of evaporation and vertical soil water movement
processes (Olioso et al., 2005). The model LWF-BROOK90.jl, a process-based, one-dimensional SVAT model, was utilized
for the analyses presented in this study. It represents a complete reimplementation of the LWF-BROOK90 model in the Julia
programming language (Bernhard et al., 2020), building on the code from the R package LWF-BROOK90R, its Fortran source
code, and the original BROOK90 (v4.8) implementation (Federer et al., 2003; Schmidt-Walter et al., 2020). The water flux is
computed by numerically solving the Richards equations using the Mualem–van Genuchten hydraulic parameters and
preferential flow (van Genuchten, 1980; Mualem, 1976) governs unsaturated and saturated water movement in soil. The model
dynamically adjusts percolation rates depending on soil hydraulic conductivity and water potential gradients.





### 2.3.1 Model input and vertical discretization

In general, the LWF-BROOK90.jl model requires as input soil hydraulic (Mualem-van Genuchten parameters, hydraulic
conductivity) and vegetation properties (e.g., leaf area index, root depth, and stomatal resistance), time series of detailed
meteorological data (precipitation, temperature, humidity, wind speed and radiation) as well as general parameters describing
the SVAT system. These data were derived from three main sources - in situ field measurements, data provided by ICOS site
operators and relevant literature (Supplement, Table S1).

For an adequate representation of the vertical variability of soil water dynamics, the soil profiles were discretized into layers
with a vertical resolution of 5 cm in the upper soil horizons (up to 30 cm) and 10 cm for deeper horizons (up to 1.7 m for
Kienhorst site and 1.2 m for Tharandt site). The layer thickness was adjusted depending on expected gradients in soil hydraulic
properties and rooting depth. This stratification allows for a more accurate simulation of water retention, percolation, and root
water uptake across the profile. Furthermore, it ensures that soil moisture dynamics are realistically captured at the defined
observation depths, allowing for a consistent comparison between simulated and observed soil moisture values.

### 190 2.3.2 Modell calibration

The objective of the calibration process was to identify inversely key soil hydraulic and vegetation-related input parameters of
the model by using soil moisture time series recorded from March 2023 to October 2024 in the four different depths within
the 11 soil profiles each. The input parameter values and ranges were derived from three main sources - in situ field
measurements, data provided by ICOS site operators and relevant literature (Supplement, Table S1). To optimize the
calibration process, a preselection of input parameters was conducted based on the results of a sensitivity analyses (Aguilar
Avila, 2024). Parameters that exhibited negligible influence on model outcomes were fixed to reduce computational
complexity (Supplement, Table S1). Finally, 17 critical input parameters on model performance were chosen for calibration
within their predefined ranges. These parameters encompassed variables related to canopy structure, vegetation hydraulics,
root distribution, soil physical processes and hydraulic properties; 11 of valid for the entire subsurface, six of them with
individual variations respecting the defined discretization scheme. For calibration, 100,000 random combinations of these 17
parameters were generated using the Latin Hypercube Sampling (LHS) strategy ensuring a well-distributed exploration of the
parameter space within a feasible range of computational costs. The LWF-BROOK90.jl model was then executed for each of
the 100,000 parameter sets. An initialization (spin-up) period of approximately three months was implemented to minimize
the influence of initial condition uncertainties on the simulation of soil moisture and water fluxes. Model performance was
evaluated using the Kling-Gupta Efficiency (KGE) score (Supplement, Section Statistics), as it provides a balanced assessment
of correlation, bias, and variability between simulated and observed soil moisture dynamics across multiple depths (Gupta et
al., 2009). For further analysis of water balance components, the 30 best-performing simulations according to KGE for each
of the 11 individual profiles per plot were selected. This selection represents a compromise between model accuracy and the
exploration of plausible model behaviour, allowing for a robust and nuanced evaluation of model uncertainty.





### 2.3.3    Model output and evaluation


In addition to the various variables relevant to water dynamics and recharge processes calculated by the model, such as evapotranspiration (ET) and root water uptake (RWU), the primary focus of the evaluation was on groundwater recharge generated at daily resolution. While the other variables provided important insights into the water balance and plant–soil interactions, groundwater recharge was of particular interest due to its critical role in sustaining long-term water availability

and its sensitivity to both model-inherent factors and the spatial variability of soil moisture used for calibration. To investigate this in detail, the 30 best-performing simulations for individual soil profiles, each based on its respective calibrated parameter set, were analysed to determine whether uncertainties arising from model structure, such as parameter interactions, equifinality, and dimensional simplifications, have a greater influence on model outputs than those related to spatial heterogeneity in soil texture and hydraulic properties within the plot. In addition, a bootstrapping procedure was performed using a two-sample

Kolmogorov–Smirnov (KS) test (Supplement, Section Statistics) to determine the minimum number of soil profiles necessary for a representative estimation of groundwater recharge across the studied 20 × 20 m plots. The analysis was based on simulated groundwater recharge values produced by the LWFBrook90 model for 11 soil profiles, each yielding 30 values derived from calibration with observed soil moisture measurements, resulting in a total of 330 values. For varying numbers of profiles (n = 1 to 11), random subsets of n profiles were repeatedly drawn (1000 iterations per subset size), and their aggregated value

distributions were statistically compared to the full dataset (all 11 profiles). The KS test was employed to assess whether the distribution of the subsample significantly differed from that of the complete set.

### 3    Results

### 3.1    Observed soil moisture dynamics

Initially, the visual examination of soil moisture time series across the 11 soil profiles each, measured at four depths (10, 30,

50, and 80 cm) at the Kienhorst (Fig. 4) and Tharandt (Fig. 5) plot, revealed distinct patterns of variability.

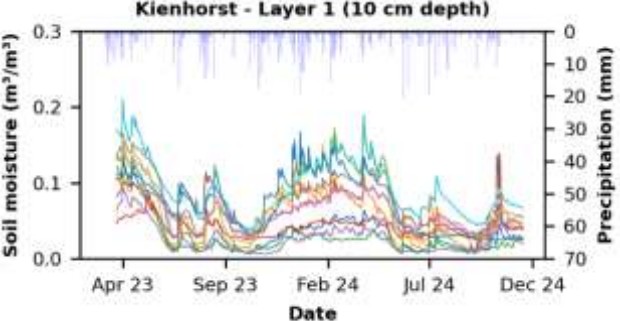

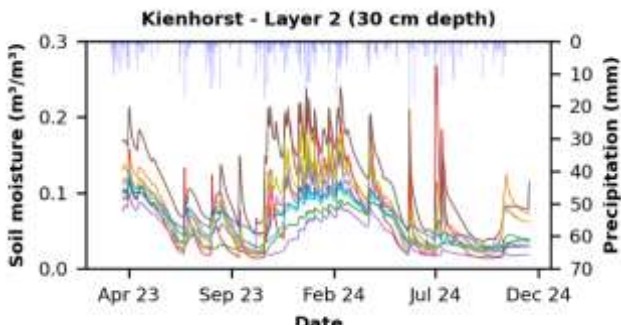





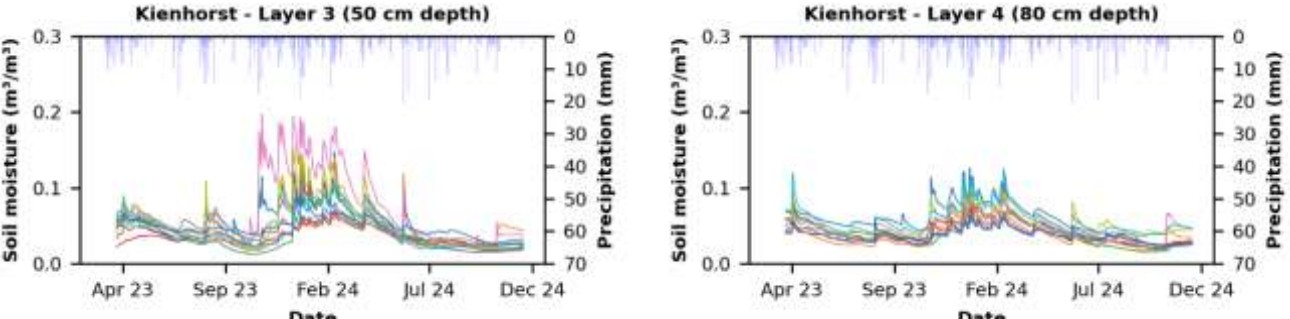

**Figure 4** Observed soil moisture at Kienhorst plot – Layer 1, 10 cm, Layer 2, 30 cm, Layer 3, 50 cm and Layer 4, 80 cm, each line represents the soil moisture of one of the 11 profiles

Seasonal effects were evident at both study locations across nearly all soil depths due to the fluctuating intensity and timing of precipitation events and the changing consumption of water by vegetation. However, the redistribution and consumption of precipitation water on its way through the unsaturated soil zone resulted in a diminished manifestation of seasonal patterns in the lower soil horizons. Notably, certain soil profiles at the Tharandt plot exhibited an almost uniform low moisture level throughout the year in the lower horizons, indicating limited seasonal variability in these depths.


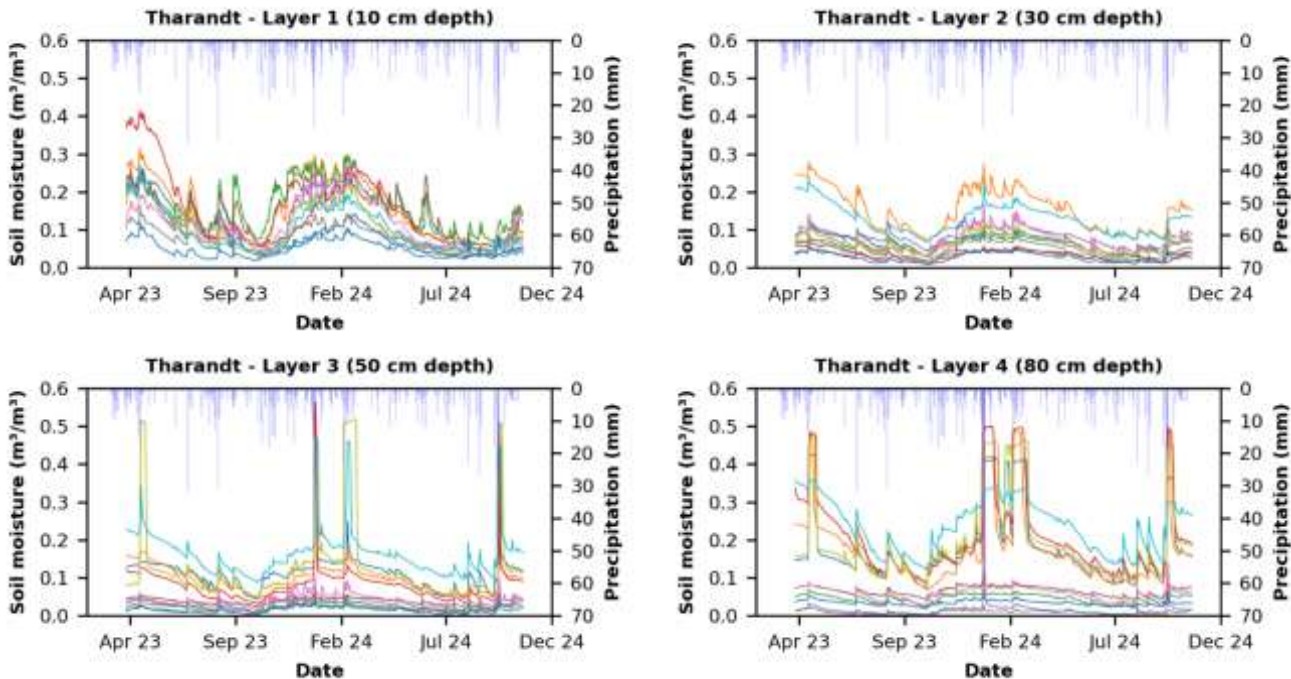

**Figure 5** Observed soil moisture at Tharandt plot - Layer 1, 10 cm, Layer 2, 30 cm, Layer 3, 50 cm and Layer 4, 80 cm, each line represents the soil moisture of one of the 11 profiles

 

Additionally, rapid and pronounced increases in soil moisture following heavy rainfall events were observed in several, but
not all, soil profiles at the Tharandt plot. At the Kienhorst plot, the range of observed soil moisture values and the variability
between individual profiles were consistently low both in dry and wet periods with evident uniformity of soil moisture across
all depths. Soil moisture exhibited a consistent pattern across all depths, with a range of approximately 15 vol. % between
minimum and maximum values. Conversely, the Tharandt plot was characterized by significant fluctuations in soil moisture
content, coupled with considerable variability between individual profiles, particularly during and following heavy
precipitation events occurring here more often. At this plot, the range between minimum and maximum moisture content
reached nearly 50 vol. %, highlighting the pronounced heterogeneity of soil water dynamics.

## 3.2      Model calibration

### 3.2.1      Simulated soil moisture dynamics and their performance

The calibration outcomes demonstrate strong correspondence between the temporal dynamics and magnitude of soil moisture
changes in the simulated compared to the observed soil moisture across most profiles at both study locations over depth, which
are reflected by constantly high KGE values (Fig. 6 and 7, for complete results of calibration see Supplement, Fig. S1 and S2).
The model calibration effectively reproduced seasonal variations evident in the measured soil moisture values, including the
response to prolonged dry periods and the rapid recovery following significant precipitation events.

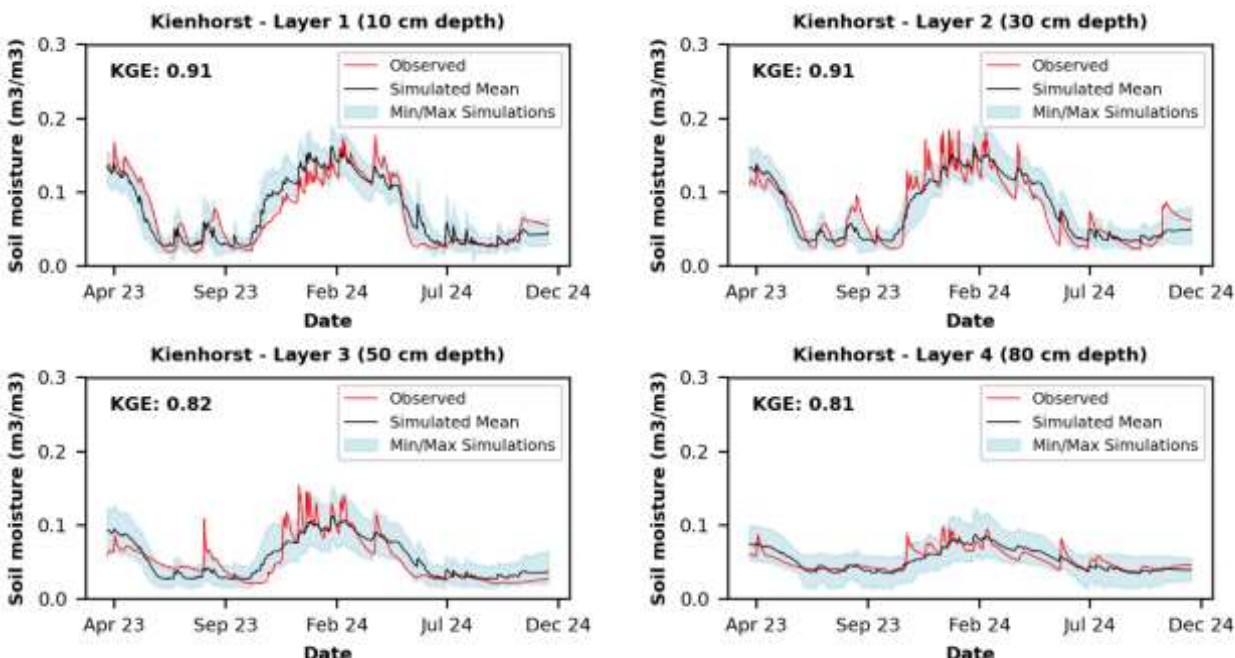

**Figure 6** Observed and simulated soil moisture exemplary for soil profile 9 at Kienhorst plot - Layer 1, 10 cm, Layer 2, 30 cm, Layer 3, 50
cm and Layer 4, 80 cm





Minor discrepancies in the absolute values, timing of peaks and magnitudes were detected in the profiles over the depth.
Significant deviations were observed here during periods of extreme wet conditions, especially at the Tharandt plot represented
by lower KGE values. Specifically, while the timing of sharp increases in soil moisture following heavy rainfall events closely
aligns with the observed data, the model underestimates the magnitude and the duration of these rises (Fig. 7, Profile 1, 50 and
80 cm depth).

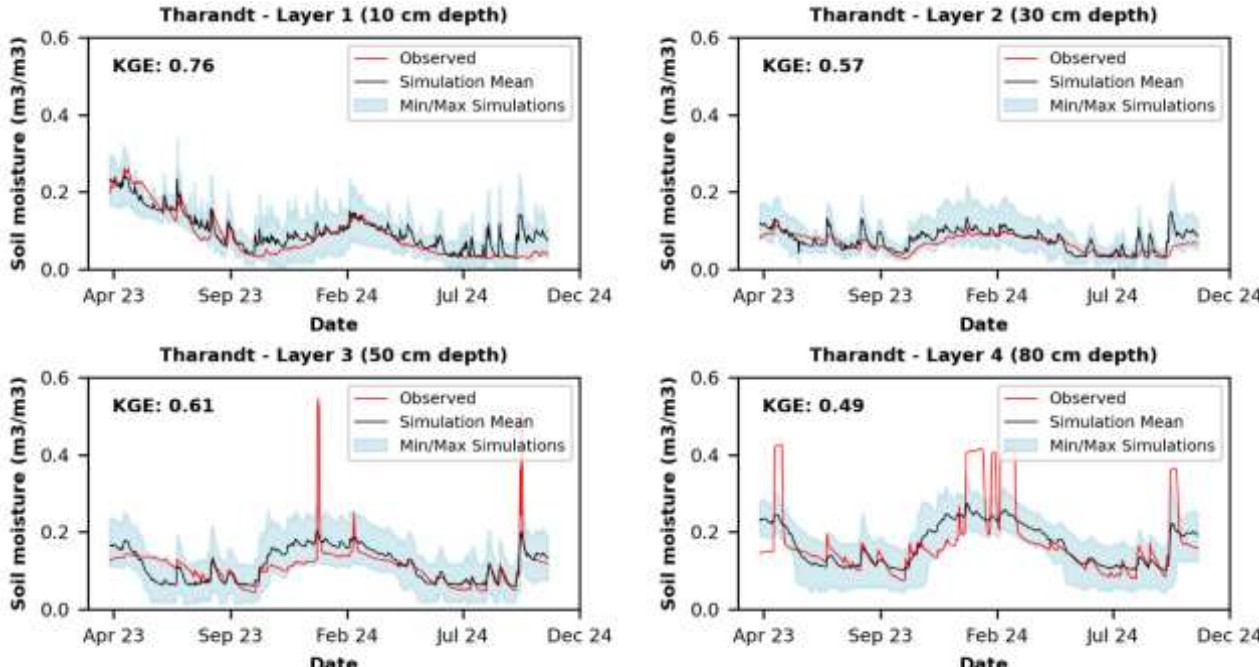

**Figure 7** Observed and simulated soil moisture exemplary for profile 1 at Tharandt plot - Layer 1, 10 cm, Layer 2, 30 cm, Layer 3, 50 cm
and Layer 4, 80 cm

Despite the existing deviations, the model performance demonstrates a high level of precision in replicating absolute soil
moisture values across the 11 soil profiles and four measured depths (10, 30, 50, and 80 cm), what can be proven by consistently
high Kling-Gupta Efficiency (KGE) scores across most profiles (Supplement, Table S2 and S3). Even for profiles with greater
variability or complex conditions, the model maintains acceptable accuracy, with KGE values not falling below 0.01.

### 3.2.2    Calibrated parameter combinations

The analysis of the variation of the individual input parameters in the 11 soil profiles each based on the 30 best simulations
derived from calibration highlights significantly greater variation in the parameters at the Tharandt plot compared to the
Kienhorst plot (Supplement, Fig. S3 and S4). Notably, the highest variation was observed in soil hydraulic and soil process



parameters, such as ths (saturated volumetric water content, thr (residual volumetric water content), ksat (saturated hydraulic conductivity), idepth (soil depth until which infiltration is distributed), qffc (quickflow fraction of infiltrating water at field capacity), drain (multiplier to partially activate drainage) and length slope (slope length for downslope flow). Especially at the Tharandt plot, also a strong variation in these parameters can be observed across the discretized soil layers, whereas the variation over depth is less pronounced at the Kienhorst plot. Increased variation of the soil hydraulic parameters at the

Tharandt plot can be attributed to the heterogeneous soil composition, which is characterised by alternating, poorly permeable layers, stones and underlying layers with low permeability. This heterogeneity increased the sensitivity of the plot to precipitation events, which is reflected in pronounced differences in soil moisture dynamics between the profiles and consequently also in the calibrated parameter sets. For the parameters that characterize the vegetation, there was greater variation for the parameters ksnvp (reduction factor to reduce snow evaporation), cvpd (vapor pressure deficit at which

stomatal conductance is halved) and maxrootdepth (maximum root depth), although these were similarly strong at both the Kienhorst and Tharandt plot.

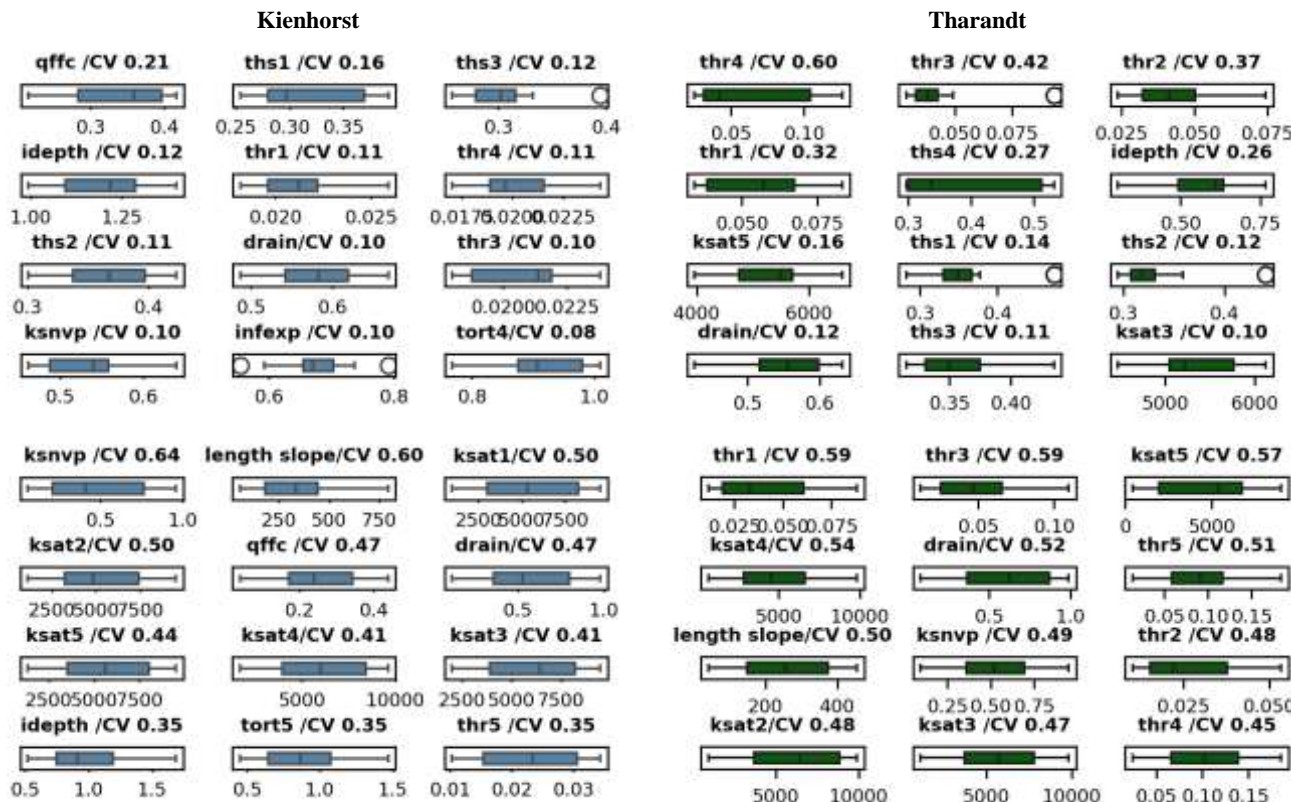

**Figure 8** Parameter variation for the 12 parameters with the highest coefficient of variation for the Kienhorst and Tharandt plot – based on the mean of the 11 profiles (upper figures) and based on the 30 best simulations for soil profile 1 (lower figures)






For further interpretation the coefficient of variation (CV) was calculated based on the mean value of the individual parameters in the 11 profiles, reflecting the uncertainty arising from the different characteristics of the 11 soil profiles at the two plots (Fig. 8) as well as based on the individual parameters of the 30 best simulations for the single profiles, reflecting model uncertainty (Fig. 8). The higher CV values observed when comparing the individual parameters across the 11 soil profiles

again highlight the greater variability in soil hydraulic and process parameters at the Tharandt plot (maximum CV 0.6 for Tharandt versus CV 0.21 for Kienhorst). This increased variation reflects the higher heterogeneity in soil characteristics at Tharandt compared to the more homogeneous conditions at Kienhorst. Further, the comparison of the two types of calculated CV allows determining whether the variability in model outcomes is driven more by differences in soil profile characteristics or by the uncertainty inherent in the model structure and parameterization. It becomes evident that the variation within the 30

best simulations for the individual soil profiles exceeds the variation observed between the 11 soil profiles for most of the parameters. This effect is particularly pronounced at the Kienhorst plot.

### 3.3. Influence of spatial variability and model parameter uncertainty on simulated groundwater recharge

Besides certain similarities, clear differences in the time series of daily groundwater recharge estimates at the two plots can be observed (Fig. 9). The daily groundwater recharge time series revealed pronounced fluctuations throughout the study periods,

with seasonal and vegetation-period effects distinctly evident. At both the Kienhorst and Tharandt plots, groundwater recharge predominantly occurs outside the vegetation period, between November and April, when the consumption of precipitation water by processes such as evaporation, root water uptake and transpiration is reduced to a minimum.

Conversely, between April and October, these processes utilize nearly all available precipitation water, effectively limiting groundwater recharge during the vegetation period. A detailed examination of the simulated time series reveals that the greatest

discrepancies between profiles, reflecting the uncertainty arising from the different characteristics of the 11 soil profiles at the sites, emerge following precipitation events occurring outside the vegetation period. During the growing season, when soil water fluxes are generally low due to high evapotranspiration, the time series exhibit notably similar patterns across profiles, indicating limited vertical percolation and groundwater recharge. Substantial differences, however, are observed in the magnitude and timing of groundwater recharge between the two plots. At Kienhorst, recharge is characterized by relatively

uniform and moderate values during the non-vegetation period. Conversely, the Tharandt plot exhibits episodic and significantly more intense recharge events, predominantly following stronger precipitation.

Considering the range of the 30 simulated time series of groundwater recharge per profile, reflecting model uncertainty, it becomes evident that the largest discrepancies between the individual simulations occur predominantly during periods outside the growing season and following heavy precipitation events (Fig. 9). Moreover, the spread between the simulations is notably

more pronounced at the Tharandt plot compared to the Kienhorst plot.





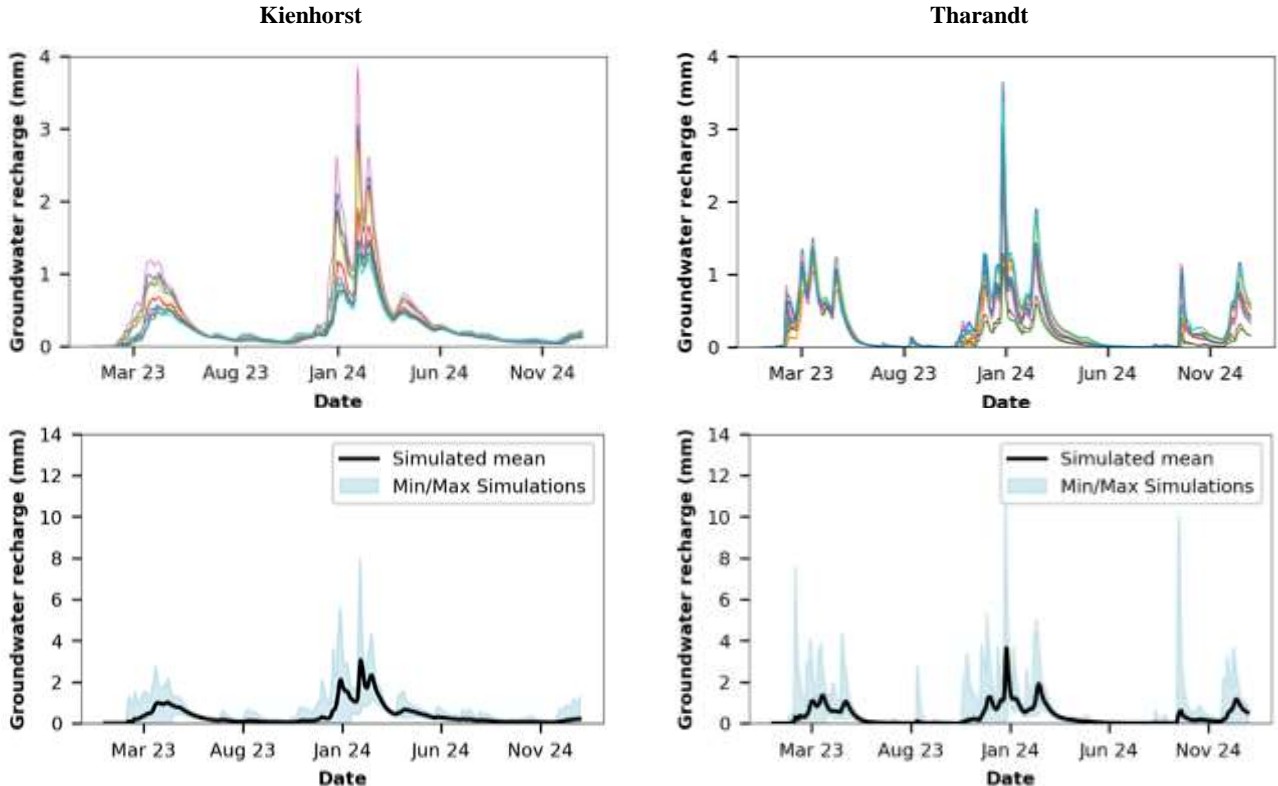

**Figure 9** Simulated time series of daily groundwater recharge based on model parameterisation of the individual 11 profiles for Kienhorst
and Tharandt plot - mean of the 30 best simulations per soil profile (upper figures), simulated time series of daily groundwater recharge –
mean as well as minimum and maximum values of the 30 best simulations exemplary for soil profile 1 (lower figures)

An analysis of cumulative groundwater recharge highlights substantial spatial variability across the 11 soil profiles at both
plots separated for the years 2023 and 2024 (Fig. 10). At the Kienhorst plot, mean annual recharge values of the 30 best
simulations for the 11 profiles ranged from 60 to 126 mm in 2023 and from 115 to 194 mm in 2024, while the groundwater
recharge calculated using the model parameterisation based on the mean value of the 11 soil moisture time series was 77 mm
for 2023 and 139 mm for 2024. At Tharandt, values varied between 80 and 134 mm in 2023 and between 34 and 123 mm in
2024, the corresponding mean values amount to 119 mm for 2023 and 103 mm for 2024.



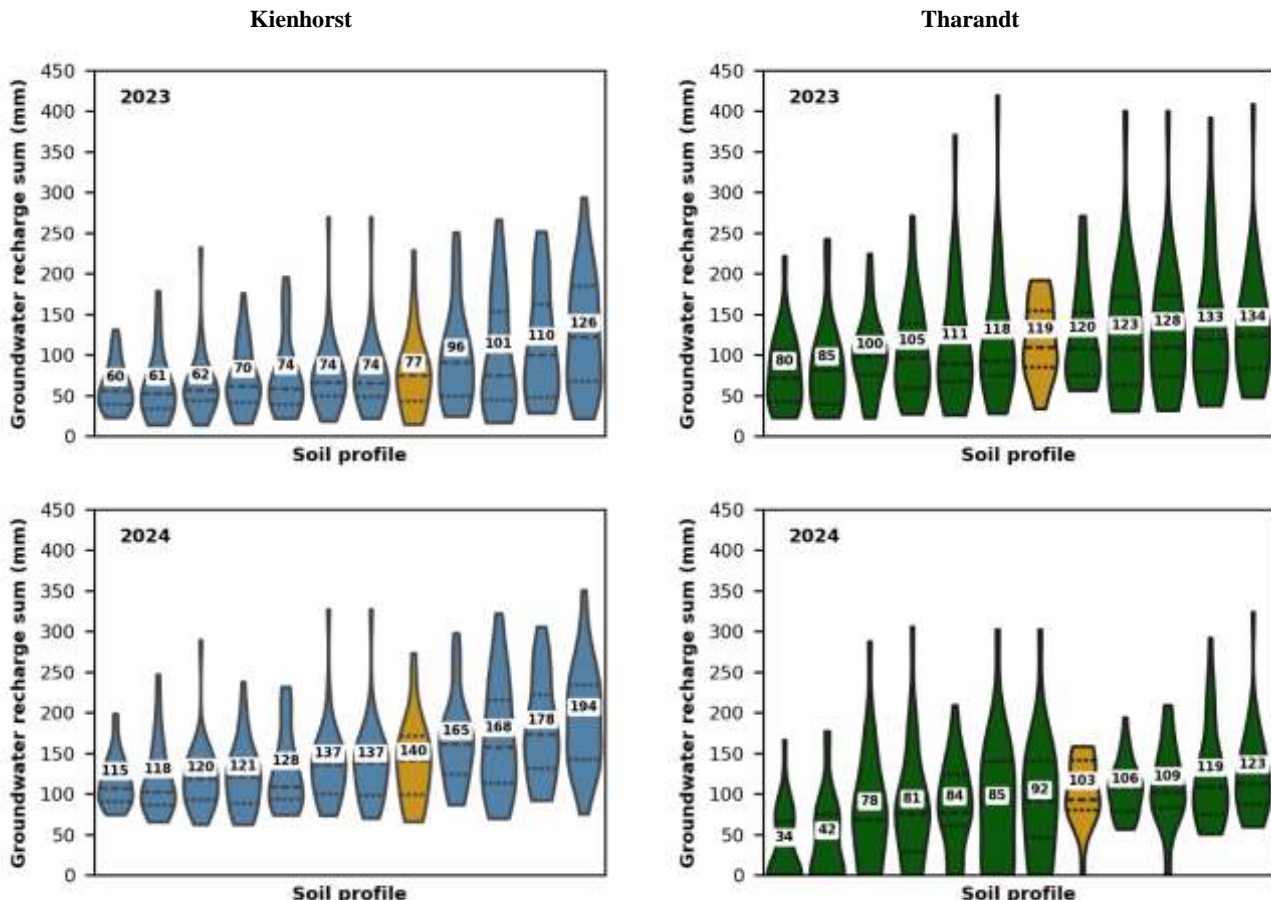

**Figure 10** Calculated cumulative groundwater recharge volume ranges based on the parameter sets of the 30 best simulations for the individual 11 soil profiles (blue/green) as well the average value given as a number for each profile - separated for 2023 (upper figures) and 2024 (lower figures) at Kienhorst and Tharandt plot, furthermore the calculated cumulative groundwater recharge volume ranges based on the parameter sets of the 30 best simulations for the mean value of the 11 soil profiles (orange), the violins display the upper bound as the third quartile (75th percentile), the lower bound as the first quartile (25th percentile) as well as the median, the outer edges of the violin extend to the actual minimum and maximum values of the data.

The results of the performed bootstrapping procedure by using a two-sample Kolmogorov–Smirnov (KS) test indicate that when using a low number of soil profiles, specifically fewer than 5 profiles for the Kienhorst and Tharandt plot (Fig. 11), significant differences from the full dataset frequently occur. In these cases, the proportion of tests showing non-significant differences remains on a lower level, indicating that such small subsets do not adequately represent the overall groundwater recharge distribution. Conversely, with more than 6 profiles at both plots, over 95% of tests indicate non-significant differences, meeting the typical criterion for statistical significance.

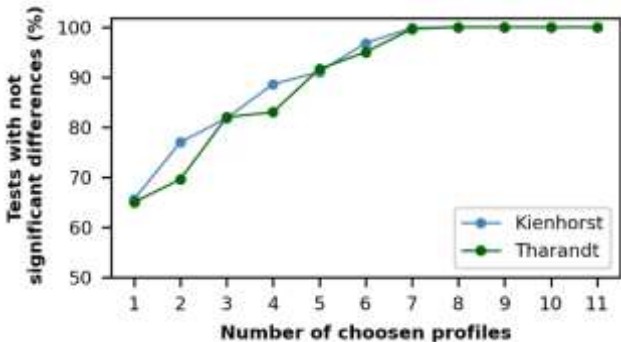

**Figure 11** Percentage of tests with not significant differences between the aggregated groundwater recharge value distributions of the chosen profiles (30 * n profiles) and the full dataset of groundwater recharge values (11 profiles * 30 = 330) for varying numbers of profiles assuming under the null hypothesis that the sample originates from the same distribution as the overall dataset.

## 4       Discussion

### 4.1      Observed soil moisture variability

Observed soil moisture at the investigated forest plots revealed pronounced differences, both laterally across the investigated area and vertically with soil depth with larger fluctuations and absolute values at the Tharandt plot. These differences are primarily driven by site-specific factors like rainfall/throughfall patterns, soil hydraulics as well as subordinated by vegetation properties. The relatively uniform temporal and absolute progression of soil moisture observed at the Kienhorst plot reflects the homogeneous characteristics of its soil matrix. The sandy soil, characterized by evenly distributed pore spaces and low

water-holding capacity over the entire soil profile, contributes to the consistent distribution of soil moisture with moderate volumetric content across all depths. In contrast, likewise to the soil moisture patterns found by Berthelin et al. (2023) the soil moisture dynamics at the Tharandt plot exhibit significant variability, attributable to its heterogeneous soil characteristics. This includes alternating layers of silty soil with variable permeability, higher stone content, and underlying geological features such as impermeable layers of clayey material beneath the upper soil horizons. The observations align with the findings of

Vereecken et al. (2016), who highlighted the influence of soil texture and hydraulic properties on spatial variability in soil moisture. At Tharandt plot, accumulation of stagnant water during periods of heavy rainfall at some locations underscores the site's sensitivity to rainfall. This phenomenon is consistent with prior studies by Famiglietti and Wood (1994), which emphasize the role of reduced soil permeability and hydraulic conductivity in limiting infiltration rates and enhancing water retention within heterogeneous soil profiles.




## 4.2    Model calibration

Despite the spatial heterogeneities in soil characteristics, the SVAT model, when calibrated with site-specific soil moisture
observations, was capable of capturing the observed dynamics and variability for most profiles at both plots with acceptable
precision. This indicates the model's capacity in general to represent key hydrological processes such as infiltration,
evapotranspiration, root water uptake, water retention and soil water redistribution under diverse conditions, what can be
proven by sufficient KGE values for most of the observation points.

Nevertheless, discrepancies between observed and simulated soil moisture remain, particularly in response to high-intensity
precipitation events followed by rapid and strong increase in soil moisture. The model results tended to underestimate the rapid
increase and struggles to replicate the absolute magnitude of soil moisture values. These discrepancies suggest limitations in
the parameterization of the model for heterogeneous soil profiles and highlight inherent simplifications in the 1D modeling
approach. This may restrict the model's capacity to fully capture the complexities of plot-specific conditions and processes
such as spatial soil heterogeneity and the lateral water flux between neighbouring soil compartments (McDonnell, 1990), which
results in an unsatisfactory simulation of stagnant and groundwater-influenced plots. The importance of focussed recharge
processes due to soil heterogeneities was also pointed out by Ries et al. (2015) and Berthelin et al. (2023). Yet, the generally
high KGE performances indicate that the weaknesses of the model in representing this behaviour well did not substantially
affect the overall realism of the simulations. In addition, we must account that the observed differences in absolute soil moisture
values may partly stem from the measurement uncertainty that originates from the accuracy of the soil moisture sensors used
in this study, which is ± 3 % of soil moisture (Truebner Company, 2025) and the effective uncertainty of field soil moisture
measurements inherent to all soil moisture measurements (Jackisch et al., 2020). This inherent variability introduces additional
uncertainty into the observed data, which should be taken into account when interpreting deviations between simulated and
observed values, particularly in profiles where differences fall within this error range.

On the other site, the analysis showed that incorporating multi-depth soil moisture observations significantly improves the
representativeness of the simulated soil water distribution, especially in heterogeneous soil systems as in Tharandt. Multiple
observation points along the soil profile allow for a more nuanced assessment of the sensitivity of individual model parameters
at different depths by calibrating the model against a higher density of data that captures both surface and subsurface processes.
This was also observed by Houska et al. (2014), who demonstrated that the inclusion of soil moisture data at different depths
increases the representativeness of the simulated soil water distribution and thus increases the identifiability of model
parameters.

## 4.3    Variability of calibrated model parameters

An analysis calibrated model parameter sets for both plots revealed a higher degree of variation in area and depth at the
Tharandt plot compared to Kienhorst plot, as indicated by the coefficient of variation (Supplement, Fig. S3 and S4). This





reflects the greater heterogeneity in soil physical properties observed at Tharandt, whereas Kienhorst is characterized by more
homogeneous plot conditions, and is a direct result of the spatially variable observed soil moisture dynamics found within the
plots. Despite this broader range of plausible parameter combinations at Tharandt plot, the overall model performance
remained at an acceptable level. However, a slightly reduced agreement between observed and simulated soil moisture values
was observed compared to Kienhorst plot, suggesting a more complex interplay between parameter uncertainty and model
behaviour in heterogeneous environments. The analysis further reveals that key soil hydraulic parameters and soil process
parameters show a greater variation compared to vegetation parameters. This underscores their key role in soil water dynamics
and highlights their contribution to overall model uncertainty. These insights are supported by the findings of Kreye and Meon
(2016), who highlighted the significant impact of sub-scale spatial variability in soil hydraulic properties on hydrological
process modelling. Similarly, Scharnagl et al. (2011) pointed out the value of incorporating prior knowledge of soil hydraulic
parameters to enhance parameter identifiability in inverse modelling approaches. Complementing these perspectives, Baroni
et al., 2010 demonstrated that uncertainties in the determination of soil hydraulic properties can substantially affect the overall
performance of hydrological models.

Furthermore, analysis of the 30 best parameter combinations for each individual soil profile revealed that model-based
uncertainty (the variation within the 30 best-performing parameter combinations) exceeds plot-based variability (differences
between the 11 soil profiles). This finding is particularly pronounced at the Kienhorst plot, where the limited variation in
measured soil properties resulted in relatively minor differences in input parameterization across profiles.

In contrast, the wide range of equally well-performing parameter sets for individual profiles reflects the issue of equifinality,
where multiple parameter combinations can yield similar simulation outcomes. This suggests that, for Kienhorst plot, model
parameter ambiguity dominates the overall uncertainty. This means that at plots with relatively uniform soil properties, where
physical variability is limited, model-based uncertainty is the more important factor for prediction accuracy. At the Tharandt
plot, by contrast, the difference between site-related and model-related parameter variation was less pronounced. For several
parameters, both sources of uncertainty, natural spatial heterogeneity and model-based calibration uncertainty, contributed
similarly to the overall variation. This can be attributed to the more pronounced heterogeneity in soil characteristics at
Tharandt, which elevates the spatial component of uncertainty and thereby partially masks the effects of equifinality. In
particular, these findings show that the dominant source of uncertainty can vary significantly depending on plot characteristics.
At homogeneous plots, model structural uncertainty and equifinality may outweigh physical variation, while at heterogeneous
sites, spatial variability in input data may dominate. Therefore, parameter selection should be guided not only by sensitivity
analysis but also by an underploting of the plot-specific balance between model and data-driven uncertainty.

Based on the findings, the calibration of soil hydraulic parameters should remain plot-specific, as their variability and influence
on model outcomes is both large and highly context-dependent. Conversely, parameters with consistently low variability across
profiles, such as many vegetation-related parameters, may be suitable for regionalization or transfer between plots, potentially
improving scalability and reliability of water balance simulations.



## 4.4 Groundwater recharge estimation

The observed differences in groundwater recharge between the Kienhorst and Tharandt plots can be attributed most to
variations in rainfall distribution, tree species and soil texture, what is influencing infiltration rates, water retention capacities
and the timing of recharge events. In addition to seasonal effects, the main differences in the time series that can be attributed
to the different soil properties. Groundwater recharge at the Kienhorst plot is relatively uniform with moderate values reflecting
the sandy soil texture, associated with high permeability and low field capacity, promoting continuous infiltration and a steady
recharge response to precipitation events. Limited water retention in these soils allows for minor recharge even during the
vegetation period, albeit at the expense of reduced water availability for plant uptake (Hillel, 2003). In contrast, the loamy and
heterogeneous soil at the Tharandt plot, with higher field capacities and lower permeability, delays percolation until antecedent
moisture conditions exceed the storage capacity of the upper soil layers. Similar to Ries et al. (2015) recharge occurred only
in distinct pulses following heavy rainfall events. Moreover, the increased water retention at Tharandt enhanced soil water
availability for vegetation, effectively suppressing groundwater recharge during the vegetation period (Hillel, 2003). These
findings underscore the critical role of soil hydraulic properties, particularly conductivity and retention capacity, in regulating
the temporal dynamics of groundwater recharge, consistent with observations by Vereecken et al. (2016) as well as the studies
of Beven and Binley (1992) and Zhao et al. (2018). This is emphasizing the complexity of water fluxes in heterogeneous soil
systems and their limiting effect on recharge efficiency.

Regarding to groundwater recharge quantities, the results fall within the range of values reported in the literature for both plots,
suggesting annual recharge rates of approximately 80–150 mm for Tharandt (Goldberg and Bernhofer, 2007) and around
100 mm for Kienhorst (Birner et al., 2015), what supports the plausibility of the model outcomes. Despite a more heterogeneous
soil and a greater variability in parameterisation at the Tharandt plot, the variability of cumulated groundwater recharge
between the different soil profiles at the more homogeneous Kienhorst plot is not substantially lower. This indicates that neither
site heterogeneity nor the range of input parameters alone fully explains the variation in recharge estimates. Rather, other
factors, such as model structure, process representation and calibration uncertainty, appear to play a decisive role in shaping
recharge variability at the catchment scale. This observation is consistent with findings by Maxwell and Condon (2016), who
emphasised the complex interplay between soil water fluxes, landscape features and recharge processes as well as stressed that
heterogeneity does not always lead to higher variability in groundwater recharge results. It further supports the notion that
model behaviour can be dominated by structural and parametric uncertainty rather than physical input variability alone. A key
contributor to this phenomenon is the concept of equifinality, as extensively discussed by Beven and Freer (2001) and Beven
(2006). While the model is able to reproduce the observed soil moisture dynamics with reasonable accuracy using different
parameter sets, the existence of multiple parameter sets with comparable performance indicates a high degree of parameter
non-uniqueness. This not only increases predictive uncertainty, but also suggests that the model structure, particularly in its
one-dimensional form, may not fully capture the spatially distributed and lateral processes that influence soil water movement
leading to uncertainties in groundwater recharge estimation.





The results further show that the calculated groundwater recharge based on the parameter sets of the 30 best simulations for the mean value of the 11 soil profiles at the Kienhorst plot represents the range of groundwater recharge of the individual 11 profiles quite well. This averaging over serval soil moisture profiles in a forest was also proven to be useful by Berthelin et al. (2023). It suggests that, in homogeneous settings, using averaged values may provide a reasonably accurate estimate. For the

Tharandt plot, the value of groundwater recharge determined in this way is closer to the maximum groundwater recharge determined for the individual 11 soil profiles. In this case, it would lead to an overestimation of the actual groundwater recharge. This highlights the importance of a differentiated consideration of plot-specific soil profiles and moisture distributions in modelling, in order to ensure realistic results. Solely relying on mean values cannot always adequately capture the system's heterogeneity and poses the risk of systematic biases in the water balance.

Hence, in order to cope with uncertainties in obtaining representative recharge estimates, it is advisable to define an adequate number of soil profiles equipped with soil moisture sensors that serve as critical calibration points. The performed bootstrapping (Fig. 11) indicated that below 6 randomly chosen profiles, spatial variability can still bias the representativeness of recharge estimates while selecting more than 6 profiles, the parameter uncertainty is the most prominent source of variability. The results demonstrate that, regardless of site-specific characteristics, the use of at least six soil profiles is

recommended to reliably capture the spatial distribution of groundwater recharge at the $20 \times 20$ m plot scale with a confidence level of 95% when using the LWFBrook90 model calibrated with soil moisture data. Nevertheless, the analysis also indicates that even when only a single soil profile is used, groundwater recharge can still be estimated with moderate confidence in approximately 50% of all test iterations. In other words, in half of the cases, the groundwater recharge derived from one soil profile matches the overall spatial distribution across the study area. In conclusion, a targeted approach using six well-

instrumented soil profiles provides a robust, efficient, and practical methodology for groundwater recharge estimation using LWFBrook90, balancing model reliability with fieldwork feasibility.

## 5    Conclusions

The findings of the investigation underscore the critical need for a sufficiently dense and vertically resolved network of soil moisture measurements to ensure robust model calibration and to constrain predictive uncertainty for estimating important

water balance components like groundwater recharge with a SVAT model. The use of mean values derived from a limited number of measurement profiles for calibration can mask extremes in soil moisture variability, leading to systematic over- or underestimations, both in homogenous and heterogeneous environments. The results of the study clearly advocate for an increased spatial coverage of soil moisture observations and modelling of groundwater recharge based on that across the analysed area, ideally, at least 6 soil profiles. Such a denser observation network would enable to cover the influence of soil

moisture variability as well as model structure and equifinality on groundwater recharge estimates. Even in areas with low observed spatial variability of soil moisture, model-based uncertainty, resulting from multiple parameter combinations yielding





similarly plausible outputs, can dominate simulation outcomes. This underlines the importance of explicitly addressing equifinality and parameter non-uniqueness in model calibration procedures.

Incorporating isotopic signatures of precipitation and soil water, in addition to soil moisture data, during calibration could be
an opportunity to reduce the negative impact of effects such as equifinality on the calibration process and the subsequent estimation of groundwater recharge. Several studies have demonstrated that the complementary use of stable isotope measurements ($\delta^{18}O$, $\delta^{2}H$) in precipitation, along with a comparison to the isotopic signatures in soil water, has proven to be particularly effective in narrowing down parameter uncertainty (Sprenger et al., 2015). By combining isotopic data with soil moisture information, a multidimensional calibration process is enabled, allowing for a more precise identification of
parameters that govern key water transport and storage processes within the soil. Furthermore, the results obtained for evapotranspiration and root water uptake, which significantly influence the estimation of groundwater recharge, can be validated through comparison with measured sap flow data. This validation step not only serves to verify the model's performance but also aids in further reducing the uncertainty in groundwater recharge estimates, thereby providing a promising direction for improving the overall reliability of future SVAT model applications.

**Code and data availability**

The source code of the LWF-BROOK90.jl model can be downloaded from https://github.com/fabern/LWFBrook90.jl
(Bernhardt, 2020). The complete model spatial input data, the meteorological data and monitoring data used in this study can be obtained upon request.

**Supplemental information**

The supplement related to this article is available online at ……………..

**Author contribution**

Conceptualization: TF, AH, MM, SS, SR; Investigation (field experiments + data collection): TF; Simulation: JA, TF; data analysis: JA, TF; Evaluation + Visualization: TF; writing (original draft preparation): TF; writing (review and editing): TF, AH, MM, SS, SR.

**Competing interests**

The contact author has declared that none of the authors has any competing interests.



**Financial support**

This work was funded by the German Federal Ministry of Food and Agriculture. (Grant number 2220WK83C4) and the Open Access Funding of the TU Dresden.

**Acknowledgements**

We would like to express our sincere thanks to the Institute of Hydrology and Meteorology at the Technical University of Dresden, in particular Thomas Grünwald, for providing access to the Tharandt site as well as the associated meteorological

data. Our thanks also go to the Landesbetrieb Forst Brandenburg, especially Alexander Russ, for granting access to the Kienhorst site and the corresponding meteorological data. Furthermore, we are grateful to Lindsey Kenyon and Benjamin Gralher for their valuable support during the field campaigns.

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
