# Peer review of "Effects of spatial soil moisture variability in forests plots on simulated groundwater recharge estimates"

_EGUsphere, 2025_

## Referee Comment (RC2)

**Review of "Effects of spatial soil moisture variability in forest plots on simulated groundwater recharge estimates"**

The manuscript investigates how the number of soil moisture measurement plots affects the parameterisation, calibration, and estimation of soil moisture and subsequently groundwater recharge (GWRCH) using the LWF-BROOK90 model, applied to two pine and spruce forest sites in Germany. The manuscript is generally well written and clearly structured. The topic itself is relevant and fits well within the scope of HESS.

However, I have substantial concerns regarding the methodological setup and the scientific and practical relevance of the presented results. Nevertheless, the study has the potential to make a meaningful contribution to both the BROOK90 modelling community and forest hydrology research. Therefore, I recommend major revisions.

**General comments:**

**Model setup and description**

- The model setup section needs further elaboration. Please provide more information about the LWF-BROOK90 model itself, not only its soil moisture component.
- It is unclear which model output was used to represent groundwater recharge (GWRCH). I assume it is the drainage from the lowest soil layer rather than outflow from the conceptual subsoil bucket with delay, but this should be explicitly stated and justified.
- It should be discussed that the model can only be used to estimate GWRCH under several restrictive assumptions: it is a 1D model with no explicit groundwater module, no lateral flow/routing, and no upward capillary flux representation.
- Please provide sources for all parameter values and parameter ranges used in the calibration. How the soil hydraulic parameters were derived for each layer (which PTF, etc.)?
- I was surprised not to find the maximum LAI parameter or its inclusion in the sensitivity analysis.
  Together with glmax, this is one of the most sensitive parameters in BROOK90. Further, the
  method used to simulate seasonal LAI dynamics (phenological scheme and parameters) is not
  described, although it strongly affects evapotranspiration and thus water availability for
  percolation.
- Information on meteorological forcing data (source, temporal resolution, spatial representativeness) is missing. If daily data were used together with standard DURATN values, this may strongly affect infiltration and percolation processes.

**Soil moisture data and calibration**

- I appreciate the substantial effort required to collect such a large soil moisture dataset. However, its application within the current modelling setup—given the research questions—raises concerns.
- You calibrated the model using the entire soil moisture dataset (~20 months), leaving no data for validation. While validation may not be the core focus, it is good (and standard) scientific practice and would strengthen the credibility of your modelling results.
- Given the large number of calibration runs, I am concerned about potential model overfitting, which may reduce model performance outside the calibration period.
- Groundwater recharge is mostly generated during winter, yet only one winter season (2023–2024) is included. This raises questions regarding the robustness of GWRCH estimates. If sensors are still operating, I strongly recommend including additional data and performing a split-sample calibration/validation.

 Please discuss how soil moisture measurement uncertainty (±3%) affects calibration results, especially during dry periods where simulated SM values often fall below 5%—resulting in possible relative errors of up to 100%.

**Missing reference simulation**

• I recommend including uncalibrated/reference simulations (using measured vegetation parameters and Mualem–van Genuchten parameters obtained from soil profile data via PTFs) for all 22 profiles. This would help to evaluate the added value of calibration for both SM and GWRCH.

**Transferability and scientific significance**

- The most critical point of the study is that the experimental setup appears highly site-specific and dependent on data availability and chosen parameters. As currently presented, the results have very limited transferability and limited practical/scientific added value.
- This should be discussed more clearly with respect to the intended audience (modellers, foresters, soil scientists, hydrologists).
- Key results should either be presented with appropriate caution or supported more robustly, especially in terms of why such a complex and expensive setup is necessary for plot-scale GWRCH estimation.

**Figures**

The quality (resolution) of all figures is currently insufficient, making them difficult to interpret.

**Scope and title**

• A substantial portion of the manuscript deals with soil moisture simulation and calibration, whereas groundwater recharge is addressed only briefly. You may wish to reconsider the title or rebalance the manuscript content.

**Specific comments**

L52: The motivation for the importance of GWRCH needs to be further elaborated. As presented in the first paragraph, it remains vague.

L76–77: This represents only one of many possible approaches.

L79–81: This is quite a bold statement and is not universally true (e.g., cases of over-calibration without validation).

L101: Why are plot-scale GWRCH estimations needed rather than multi-site or gridded estimations, which are typically required for practical applications by stakeholders?

L115: Please add the Latin names of species, full soil texture classifications, and the averaging period for the meteorological data. Additionally, include soil profile data in the Appendix to assess heterogeneity.

L135–136: If stagnant conditions at this site are well known due to shallow bedrock, what GWRCH can realistically be expected, and how is this addressed in the model setup?

L144: Do you mean ±3%?

L149–150: Please elaborate on the calibration process in more detail.

L165: Add the north direction and satellite/aerial background imagery to better assess spatial beterogeneity

L193: It is unclear where the parameter calibration ranges come from. For example, cvpd values below 1–1.5 result in unreliable transpiration and should be justified. The same applies to other sensitive parameters as well.

L195: Please include a sensitivity analysis in the Appendix and specify its boundary conditions and variables of interest.

L203: A warm-up period of only 3 months is, in my experience, insufficient for this model setup. A minimum of 6 months (preferably 1 year) is recommended. Please either extend the warm-up period or demonstrate that a longer warm-up does not significantly affect soil moisture at the start of spring 2023.

L220: This setup violates one of the core assumptions of the KS test—independence of samples—since the bootstrapped subset contains data from the full set. You should choose a different metric or modify the setup.

L230: How were raw observations post-processed? Were sensor issues, spikes, or errors encountered? (e.g., see Kienhorst L1–10 cm, red straight line after 23 Sept.)

L245: The spikes observed in Tharandt—do they indicate sensor errors or preferential/bypass flow in some plots?

L260 (270): What exactly is meant by "simulations" here—an average of the 30 'best' simulations and their uncertainty bandwidth?

L263: There appears to be a substantial underestimation of variance. Please provide the full KGE decomposition to better assess this and discuss it, as it is crucial for estimating downward water flows and, consequently, GWRCH.

L289: To my knowledge, ksnvp is one of the least sensitive parameters in the BROOK90 model, as snow evaporation rarely exceeds 1% of annual precipitation. It is therefore unclear why it is included in the calibration list.

L300: It is unclear whether the CV values represent summarized CV values for all parameters per site. Please explain this in more detail.

L301: The increased variation reflects heterogeneity in calibrated soil parameters within the model setup, not actual soil properties (unless laboratory verification was performed).

L325–326: Please discuss these findings in the context of preferential flow (bypass flow) and how it is parameterized in the model.

L334–340: The cumulative GWRCH values seem much higher than expected for the studied sites. Please include a table or plot in the Appendix showing the monthly sums of all water balance components (as you are not covering full years). Also, relate GWRCH to seasonal/annual precipitation as a percentage.

L368–386: You do not demonstrate that the calibrated model reliably estimates all water balance components—only soil moisture at certain depths.

L396–398: I disagree with this statement. It must be supported with model evidence. In many profiles, the model fails to reproduce high soil moisture peaks (likely from intense precipitation events). These peaks may represent preferential flow, which could be a major source of GWRCH (e.g. for Tharandt). Failure to simulate them may lead to significant underestimation of GWRCH during such events.

L419–420: Given the model setup, emphasis on soil moisture, and limited vegetation parameters, the results are unsurprising.

L427–429: This could become a valuable result if you analyze and discuss it in terms of the parameter list, parameter ranges, and degrees of freedom allowed during calibration. Would the conclusions change if different parameters or wider/narrower ranges were used?

L464–466: The referenced GWRCH values are not reliable. I could not find any GWRCH data for Tharandt in Goldberg and Bernhofer (2007), and the Kienhorst values are taken from ArcEGMO simulations, which are often known to underestimate ET and therefore overestimate water available for GWRCH.